# The Related Mechanisms Predicted through Network-Based Pharmacological Analysis and the Anti-Inflammatory Effects of *Fraxinus rhynchophylla* Hance Bark on Contact Dermatitis in Mice

**DOI:** 10.3390/ijms24076091

**Published:** 2023-03-23

**Authors:** Sura Kim, Ji-Hyo Lyu, Beodeul Yang, Soyeon Kim, Jung-Hoon Kim, Hyungwoo Kim, Suin Cho

**Affiliations:** 1Division of Pharmacology, School of Korean Medicine, Pusan National University, Yangsan 50612, Gyeongnam, Republic of Korea; 2Department of Microbiology, Medicine School of Jeonbuk National University, Jeonju 54907, Jeonbuk, Republic of Korea; 3Research Institute for Korean Medicine, Pusan National University, Yangsan 50612, Gyeongnam, Republic of Korea; 4Herbal Medicine Resources Research Center, Korea Institute of Oriental Medicine, Naju 58245, Jeonnam, Republic of Korea

**Keywords:** *Fraxinus rhynchophylla*, traditional medicine, inflammation, skin, dermatosis

## Abstract

*Fraxinus rhynchophylla* Hance bark has been used to treat patients with inflammatory or purulent skin diseases in China, Japan, and Korea. This study was undertaken to determine the mechanism responsible for the effects of *F. rhynchophylla* and whether it has a therapeutic effect in mice with contact dermatitis (CD). In this study, the active compounds in *F. rhynchophylla*, their targets, and target gene information for inflammatory dermatosis were investigated using network-based pharmacological analysis. Docking analysis was conducted using AutoDock Vina. In addition, the therapeutic effect of an ethanolic extract of *F. rhynchophylla* (EEFR) on skin lesions and its inhibitory effects on histopathological abnormalities, inflammatory cytokines, and chemokines were evaluated. Finally, its inhibitory effects on the nuclear factor-kappa B (NF-κB) and mitogen-activated protein kinase (MAPK) signalling pathways were observed in RAW 264.7 cells. In our results, seven active compounds were identified in *F. rhynchophylla*, and six were associated with seven genes associated with inflammatory dermatosis and exhibited a strong binding affinity (<−6 kcal/mol) to prostaglandin G/H synthase 2 (PTGS2). In a murine 1-fluoro-2,4-dinitrobenzene (DNFB) model, topical EEFR ameliorated the surface symptoms of CD and histopathological abnormalities. EEFR also reduced the levels of tumour necrosis factor (TNF)-α, interferon (IFN)-γ, interleukin (IL)-6, and monocyte chemotactic protein (MCP)-1 in inflamed tissues and inhibited PTGS2, the nuclear translocation of NF-κB (p65), and the activation of c-Jun N-terminal kinases (JNK) in RAW 264.7 cells. In conclusion, the bark of *F. rhynchophylla* has potential use as a therapeutic or cosmetic agent, and the mechanism responsible for its effects involves the suppression of inflammatory mediators, nuclear factor of kappa light polypeptide gene enhancer in B-cells inhibitor (IκB)-α degradation, the nuclear translocation of NF-κB, and JNK phosphorylation.

## 1. Introduction

*Fraxinus rhynchophylla* Hance (family Oleaceae) is a deciduous tree native to Mǎnzhōu (northeast China) and the Korean peninsula, and its bark has been used to treat patients with inflammatory or purulent skin diseases in China, Japan, and Korea [1]. *F. rhynchophylla* bark is a ‘heat-clearing herb’ according to the theory of traditional medicine that reduces heat and has a bitter taste and a cold attribute [2]. *F. rhynchophylla* has been reported to have hepato-protective [3], anti-diabetic [4], and anti-oxidative effects [5], and esculin and esculetin isolated from *F. rhynchophylla* has been reported to inhibit melanin production by regulating the activities of melanogenic enzymes [6].

Contact dermatitis (CD) can affect people of all ages and can be caused by a variety of substances, including cosmetics, jewelry, plants, chemicals, and metals. The symptoms of CD typically include redness, itching, swelling, and blisters at the site of contact. In some cases, the skin may become dry, cracked, and scaly [7]. CD patients that suffer from these symptoms often exhibit stress or suffer from other illnesses [8]. Currently, work-related dermatosis accounts for about 50% of occupational disorders and results in considerable socio-economic losses.

The standard treatment for CD usually involves avoiding the substance that triggered the reaction and using topical medications such as hydrocortisone-containing ointments to relieve the symptoms. However, repeated corticosteroid use can induce various physical and psychological adverse effects. Thus, novel therapeutic agents are required, and complementary and alternative medicines based on natural substances offer relatively safe alternatives [9,10].

Several database platforms such as the TCM-ID (Traditional Chinese Medicine Information Database), the TCM Database@Taiwan DB, TCMGeneDIT, TCMSP (Traditional Chinese Medicine Systems Pharmacology Database and Analysis Platform, TarNet, and TM-MC (a database of medicinal materials and chemical compounds used in Northeast Asian traditional medicine) have been developed to facilitate searches for active natural compounds [11]. Of these platforms, the TCMSP provides a wealth of information on herbal ingredients and compounds, therapeutic targets, related diseases, and ADME (absorption, distribution, metabolism, and excretion) properties [12]. On the other hand, DisGeNET provides information about human disease-associated genes and variants [13].

In this study, TCMSP and DisGeNET were used to confirm the relationships between inflammatory dermatosis and *F. rhynchophylla* and its component compounds. The therapeutic and anti-inflammatory efficacies of *F. rhynchophylla* were investigated in a murine model of CD and a murine macrophage cell line (RAW 264.7 cells). To accomplish this, we evaluated the effects of a 70% ethanolic extract of *F. rhynchophylla* (EEFR) on the surface symptoms of CD, including skin color and thickness, and on histopathological abnormalities and inflammatory cytokines levels in mice with 1-fluoro-2,4-dinitrobenzene (DNFB)-induced CD. In addition, the inhibitory effects of EEFR on the prostaglandin G/H synthase 2 (PTGS2; cyclooxygenase-2 (COX-2)), nuclear factor kappa-light-chain-enhancer of activated B cells (NF-κB), and mitogen-activated protein kinase (MAPK) signalling pathways were investigated.

## 2. Result

### 2.1. Seven Active Compounds Were Found in F. rhynchophylla

In total, 21 compounds were found (Appendix A), and seven satisfied the ADME criteria as active compounds (Table 1).

### 2.2. Forty-Seven Related Targets Were Found in F. rhynchophylla

Targets of the seven compounds were checked using the TCMSP database. The compounds were found to have 47 targets and 70 interactions. Of the seven compounds, β-sitosterol was the most associated with target genes (37 genes), followed by sitogluside (17 genes), 8-hydroxypinoresinol (six genes), fraxin (five genes), esculin (two genes), and sinapaldehyde glucoside (two genes). In addition, six of the active compounds targeted COX-2 (Figure 1).

### 2.3. Seven Target Genes Associated with Inflammatory Dermatosis Were Found

To confirm the relationship between the seven active compounds and inflammatory-skin-disease-related genes, we searched for inflammatory dermatosis (UMLS CUI: C3875321)-related genes in the DisGeNET platform. In total, 382 genes were found, and 368 were confirmed. The remaining 14 genes were filtered out because they were not identified by UniProt. The intersections of the target genes and the genes associated with inflammatory dermatosis were confirmed using Cytoscape 3.9.1. Seven common genes were revealed, namely, cholinergic receptor muscarinic 3 (CHRM3), 5-hydroxytryptamine receptor 2A (HTR2A), Jun proto-oncogene, AP-1 transcription factor subunit (JUN), phosphatidylinositol-4,5-bisphosphate 3-kinase catalytic subunit gamma (PIK3CG), prostaglandin G/H synthase 2(PTGS2 (COX-2)), solute carrier family 6 member 4 (SLC6A4), and transforming growth factor beta 1 (TGFB1) (Figure 2).

### 2.4. All Six Active Compounds Showed Strong Binding Affinity with PTGS2

The binding affinities between PTGS2 and the six active compounds were predicted with molecular docking analysis (Table 2). These six compounds showed strong binding affinity with PTGS2 with binding energies of less than −6 kcal/mol. The interactions between the six compounds and PTGS2 are shown in Figure 3.

### 2.5. EEFR Ameliorated Surface Symptoms and Inhibited Skin Thickening in CD Mice

Repeated stimulation with DNFB induced the skin lesions of CD such as scaling, excoriation, erythema, and skin roughness (Figure 4A,B), and these symptoms were alleviated by EEFR (Figure 4A). EEFR (600 μg/day) significantly reduced skin surface scores (Figure 4B). The mean skin thickness was significantly higher in the CTL group than in the NOR group, and EEFR significantly prevented skin thickening (Figure 4C). DNFB stimulation elevated erythema indices significantly in the CTL group, and the topical application of EEFR at 600 μg/day significantly reduced erythema indices (Figure 4D). Mice in all experimental groups had similar melanin indices (Figure 4E).

### 2.6. EEFR Prevented DNFB-Induced Histopathological Abnormalities

DNFB stimulation induced remarkable epidermal hyperplasia, hyperkeratosis, and immune cell infiltration (Figure 5A). Treatment with 600 μg/day EEFR reduced epidermal hyperplasia significantly (Figure 5B). Large numbers of infiltrating immune cells were observed around the dermis and blood vessels, and this phenomenon was also significantly reduced by EEFR treatment at 180 or 600 μg/day (Figure 5C).

### 2.7. EEFR Reduced DNFB-Induced Increases in TNF-α, IFN-γ, IL-6, and MCP-1 Levels in Inflamed Tissues

TNF-α, IFN-γ, IL-6, and MCP-1 levels were significantly increased in the tissues of mice in the CTL group. Topical EEFR application at 600 μg/day effectively reduced these DNFB-induced increases in inflammatory cytokine and chemokine levels. However, EEFR at 60 μg/day did not affect the DNFB-induced increases in TNF-α and IFN-γ levels (Figure 6).

### 2.8. EEFR Suppressed Lipopolysaccharide (LPS)-Induced Increases in Pro-Inflammatory Cytokine Expression and COX-2, and the LPS-Induced Activation of NF-κB and JNK in RAW 264.7 Cells

LPS increased the mRNA expression of TNF-α, IL1-β, IL-6, and IL-8 in RAW 264.7 cells, and these increases were reduced by pre-treatment with EEFR. In addition, COX-2 expression was also reduced by EEFR (Figure 7A). As shown in Figure 7B,C, LPS treatment resulted in the degradation of nuclear factor of kappa light polypeptide gene enhancer in B-cells inhibitor (IκB)-α in the cytoplasm and increased p65 (the active form of NF-κB) levels in the nuclear fractions. These effects of LPS were effectively blocked by EEFR treatment. Furthermore, EEFR suppressed the LPS-induced phosphorylation of JNK (Figure 7D).

## 3. Discussion

Network-based pharmacology and molecular docking analysis are now frequently used to predict the efficacies of herbal medicines and formulas and have considerably reduced the time, effort, and cost of research in this area. Accordingly, we used network-based pharmacological analysis to identify compounds with the potential to replace corticosteroids.

Using this approach, we identified seven active compounds in *F. rhynchophylla* (β-sitosterol, sitogluside, 8-hydroxypinoresinol, fraxin, esculin, sinapaldehyde glucoside, and (2S,3R,4S,5S,6R)-2-[4-[(3R,3aS,6S,6aR)-3a-hydroxy-6-(4-hydroxy-3-methoxyphenyl)-3,4,6,6a-tetrahydro-1H-furo[3,4-c]furan-3-yl]-2-methoxyphenoxy]-6-(hydroxymethyl)oxane-3,4,5-triol) and seven target genes (CHRM3, HTR2A, JUN, PIK3CG, SLC6A4, TGFB1, and PTGS2). We suggest these seven target genes might serve as markers of the activity of *F. rhynchophylla* alone or containing formulae.

In silico-determined binding affinities provide indications of the strengths of interactions between molecules. In the present study, AutoDock Vina docking scores were used to assess the binding affinities, according to which a score of <−5 kcal/mol indicates stable binding [14]. Of the seven target genes, PTGS2 (COX-2; PDB-ID: 5f19; a major inflammatory mediator) showed the highest association with the six active compounds, and thus, we used molecular docking analysis to predict the binding affinities between PTGS2 and these compounds. The results obtained confirmed that all six active compounds stably bound to PTGS2 (Table 2 and Figure 3), which suggested they act as anti-inflammatories by reducing the activity of COX-2.

Acute CD is characterized by itching, vesicles, erythema, and scaling, whereas longstanding lesions are typified by skin thickening with lichenification, pigmentation, and scaling [7,15], and in the present study, the repeated application of EEFR ameliorated scaling, erythema, and skin thickness increases in the DNFB-induced mouse model of CD (Figure 4).

The most common histological feature of acute CD is epidermal hyperplasia. In addition, immune cell infiltration has been reported in the dermis and epidermis of CD patients [16]. In the present study, EEFR inhibited epidermal hyperplasia, immune cell infiltration, and hyperkeratosis (Figure 5). Considering the results shown in Figure 4 and Figure 5, these findings imply that EEFR can prevent skin thickening by inhibiting epidermal hyperplasia and ameliorate scaling by suppressing hyperkeratosis. However, EEFR had no observable effect on melanin indices (Figure 4E), indicating it does not reduce hyperpigmentation, which is a problem associated with chronic dermatitis.

The pro-inflammatory cytokine TNF-α can stimulate keratinocytes and cause hyperplasia and hyperkeratosis in the epidermis and also influence the progression of inflammation, especially that associated with immune cells, by inducing the expression of adhesion molecules on the surfaces of endothelial cells and keratinocytes [17]. IFN-γ also contributes to the inflammatory response and with TNF-α can activate keratinocytes and T-cells and promote Th1 shifted reactions [18]. Moreover, IFN-γ directly induces immune cell infiltration into epidermal tissue, epidermal hyperplasia, and hyperkeratosis [19]. IL-6 signalling is related to inflammatory infiltration by controlling the expressions of inflammatory chemokines and adhesion molecules [20] and has been reported to promote keratinocyte proliferation and lead to inflammatory skin disorders [21]. On the other hand, MCP-1, an important mediator of immune-mediated skin diseases, can accelerate monocyte and macrophage migration and infiltration [22].

In our study, EEFR effectively lowered TNF-α, IFN-γ, IL-6, and MCP-1 levels in inflamed mouse tissues (Figure 6) and in RAW 264.7 cells (Figure 7A). In particular, EEFR effectively inhibited epidermal hyperplasia and hyperkeratosis (Figure 5). These findings suggest that EEFR ameliorates epidermal hyperplasia and hyperkeratosis by inhibiting the production of pro-inflammatory cytokines and chemokines through the prevention of keratinocyte accumulation. In addition, EEFR significantly prevented immune cell infiltration (Figure 5C), which we believe was due to its inhibitory effects on TNF-α, IL-6, and MCP-1 production. Taken together, these results imply that EEFR can suppress the production of TNF-α, IFN-γ, IL-6, and MCP-1, and thus, inhibit epidermal hyperplasia, hyperkeratosis, and immune cell infiltration. These histopathological changes finally improved the skin symptoms of CD and suppressed skin thickening.

As shown in Table 2 and Figure 3, COX-2 showed the highest association with the six active compounds of *F. rhynchophylla.* Our PCR results showed EEFR significantly suppressed LPS-induced COX-2 overexpression (Figure 7B). These results mean that the anti-inflammatory mechanism of EEFR is related to the inhibition of the COX-2 pathway.

The NF-κB and MAPK signalling pathways play central roles in cytokine production and can be activated by inflammatory cytokines and chemokines produced by various immune cells [23]. Recently, Silvia et al. reported that the ERK, p38, and JNK signalling pathways can induce intercellular adhesion molecule (ICAM)-1 expression in keratinocytes [24]. Our results showed that EEFR prevented LPS-induced IκB-α degradation in the cytoplasm, nuclear p65 accumulation, and JNK phosphorylation in RAW 264.7 cells (Figure 7D) and inhibited TNF-α induced ICAM-1 expression in keratinocytes (Appendix A). These observations suggest that the anti-inflammatory mechanisms initiated by EEFR are involved in the regulation of ICAM-1 expression via inhibition of the NF-κB and JNK pathways.

Classically activated (M1) macrophages play a role in direct damage to the epidermis and recruitment of other inflammatory cells and are found in large numbers in allergic CD skin lesions [25]. In addition, dermal macrophages are key modulators in contact hypersensitivity responses [26]. RAW 264.7 cells, a mouse macrophage cell line, used in in vitro study are commonly used in research studies to investigate the immune and inflammatory responses. While the relationship between macrophages and CD is complex and not yet fully understood, this means that RAW 264.7 cells can be a useful tool for investigating the immune and inflammatory responses for identifying potential therapeutic targets for the treatment of CD.

Recently, esculetin from *F. rhynchophylla* was reported to attenuate atopic skin inflammation induced by the house dust mite or 2,4-dinitrochlorobenzene [27]. In addition, Chen et al. reported that esculetin ameliorated psoriasis-like skin disease in mice [28]. We found that esculin, one of the standard materials of EEFR, did not affect skin lesion thickness or color in CD mice (Appendix A). This means that the anti-inflammatory efficacy of *F. rhynchophylla* is closely related to the active ingredients such as esculetin rather than esculin, and also means that additional in vitro or in vivo studies are needed to confirm the results predicted by the in silico analysis.

As shown in Figure 4, dexamethasone (DEX) did not improve skin surface symptoms including the erythema index, but significantly reduced the skin thickness. This phenomenon is thought to be a result of corticosteroid-induced skin atrophy [29]. In addition, topical corticosteroids play a limited role in the treatment of CD. In our previous study using DNFB animal models, DEX did not improve skin symptoms and significantly reduced the skin thickness [30].

The spleen/body weight ratio is considered an indicator of systemic immune function [31], and in some cases, spleen shrinkage is considered an indicator of general immune suppression. We confirmed DEX reduced the spleen/body weight ratios in CD mice and that EEFR had no effect (Appendix A), which implies that the anti-inflammatory mechanism of EEFR differs from that of corticosteroids, particularly in terms of systemic immune suppression.

## 4. Materials and Methods

### 4.1. Network-Based Pharmacological Analysis of F. rhynchophylla

#### 4.1.1. Screening of Active Compounds in *F. rhynchophylla* and Target Analysis

Compounds in *F. rhynchophylla* were confirmed using TCMSP (https://old.tcmsp-e.com/tcmsp.php (accessed on 11 January 2022)), which is widely used in network pharmacological analysis. The ADME properties oral bioavailability (OB) and drug likeness (DL) were used for active compound screening. The following criteria were used: OB ≥ 20 (%) and DL ≥ 0.18. TCMSP was used to verify the target information, and the DisGeNET platform (https://www.disgenet.org (accessed on 12 January 2022)) was used to identify inflammatory dermatosis-related genes. The official gene names of the targets were checked using UniProt (http://www.uniprot.org/ (accessed on 17 January 2022)) [32].

#### 4.1.2. Network Analysis

Cytoscape 3.9.1 (https://cytoscape.org/ (accessed on 19 January 2022)) was used to confirm the compound–target and inflammatory-dermatosis–target networks [33].

#### 4.1.3. Docking Analysis

Docking analysis was performed as described previously [34], and the binding affinities of the active compounds and target proteins were examined using AutoDock Vina (The Scripps Research Institute, La Jolla, CA, USA) [35].

### 4.2. In Vivo and In Vitro Experiments

#### 4.2.1. Preparation of *F. rhynchophylla* Bark Extract

*F. rhynchophylla* bark was purchased from the Kwangmyungdang company (Ulsan, Korea), and a specimen was deposited in the School of Korean Medicine, Pusan National University (Voucher no. MS2017-0014). The *F. rhynchophylla* bark was authenticated by Professor Jung-Hoon Kim.

The dried bark of *F. rhynchophylla* (200 g) was extracted using a standard method, as we previously described [30], and yielded 6.5 g of lyophilized powder (EEFR, yield 3.25%). A sample of EEFR was also deposited in the School of Korean Medicine, Pusan National University (Voucher no. MH2017-0014). The fingerprint of EEFR is provided in the Appendix A.

#### 4.2.2. Animals

Male 6-week-old Balb/c mice were purchased from Samtaco (Osan, Republic of Korea). The mice were housed under specific pathogen-free conditions under a 12 h light/dark cycle with free access to standard rodent food and water. All animal experiments were approved beforehand by the animal care and use committee of Pusan National University and conducted according to institutional guidelines (PNU-2015-0979).

#### 4.2.3. Induction of CD and Experimental Design

CD was induced using DNFB in Balb/c mice, as we previously described [30]. Briefly, the mice were randomly divided into six groups, namely, a normal (NOR) group of non-treated mice (*n* = 6); a control (CTL) group of non-treated CD mice (*n* = 8); three EEFR groups of CD mice treated with 60, 180, or 600 μg/day of EEFR for six consecutive days (*n* = 8/group); and a dexamethasone (DEX)-treated CD group treated with 150 μg/day of DEX for six consecutive days (*n* = 8). EEFR and DEX were dissolved in 70% ethanol and diluted in vehicle (AOO, acetone: olive oil, 4:1). The EEFR and DEX solutions were applied to the shaved backs of Balb/c mice. The experimental schedule is summarized in the Appendix A).

#### 4.2.4. Skin Surface Scores and Thickness Measurements

Skin lesions were observed using a digital camera (Olympus, Tokyo, Japan). Skin surface scores were assessed by summing the skin lesion scores for scaling, excoriation, erythema, and skin roughness. The scores were evaluated using a modified version of Amano’s method [36]. Skin tissues were cut into 5 mm diameter pieces, and the thicknesses were measured using vernier calipers (Mitutoyo, Kanagawa, Japan).

#### 4.2.5. Erythema and Melanin Indices

The erythema and melanin indices were determined using dermo-spectrophotometer measurements (Cortex Technology, Hadsund, Denmark) obtained at three different points on the skin surface per mouse.

#### 4.2.6. Histopathological Examinations

Skin tissues were fixed in 10% (*v*/*v*) formaldehyde, embedded in paraffin, sectioned at 4 μm, and stained with hematoxylin and eosin. The stained tissue slides were observed for histological changes under a light microscope (Carl Zeiss AG, Oberkochen, Germany) at ×100.

#### 4.2.7. Evaluations of Epidermal Hyperplasia and Immune Cell Infiltration

To evaluate epidermal hyperplasia, the vertical distances between the basal lamina and the outer stratum granulosum were measured. Three random measurements were made per slide using the Zen program (ZEIZZ, Jena, Germany). To evaluate immune cell infiltration, cells were enumerated using a cell-counting grid (1024 × 1024 μm) in four randomly selected, non-overlapping regions per slide. Immune cells were defined as macrophages, polymorphonuclear leukocytes, lymphocytes, eosinophils, plasma cells, and giant cells [37].

#### 4.2.8. Inflammatory Cytokine and Chemokine Levels

Skin tissues (30 mg) were lysed in 300 μL PRO-PREP protein solution (iNtRON, Seoul, Korea) using a bullet blender (Next Advance, Averill Park, NY, USA). The levels of tumour necrosis factor (TNF)-α, interferon (IFN)-γ, interleukin (IL)-6, and monocyte chemoattractant protein (MCP)-1 in the lysates (50 μg) were determined using the cytometric bead array (CBA) method (B.D. Biosciences, San Diego, CA, USA).

#### 4.2.9. Isolation of Total RNA and Reverse Transcription Polymerase Chain Reaction (RT-PCR)

The total RNA was extracted from cells using TRIzol (Invitrogen, Carlsbad, CA, USA). Complementary DNA (cDNA) was synthesized as described previously [38]. The PCR cycling conditions were 95 °C for 5 min, followed by 20–28 cycles of 95 °C for 30 s, 55 °C for 30 s, and 72 °C for 40 s. The primer sets used are shown in Appendix A. The target gene expressions were normalized versus GAPDH.

#### 4.2.10. Western Blot Analysis

Protein isolation and Western blotting were performed as previously described [36]. Blots were developed using an enhanced chemiluminescence system (SuperSignal^®^ West Femto, Thermo Scientific, Rockford, IL, USA).

#### 4.2.11. Statistical Analysis

Data were analysed using Kruskal–Wallis tests followed by Dunn’s comparison test. Prime 5 for Windows version 5.01 (GraphPad Software Inc., La Jolla, CA, USA) was used for the analysis. The results are presented as means ± standard deviations, and statistical significance was accepted for *p* values < 0.05.

## 5. Conclusions

This study identifies six active compounds (β-sitosterol, sitogluside, 8-hydroxypinoresinol, fraxin, esculin, and sinapaldehyde glucoside) and seven target genes (CHRM3, HTR2A, JUN, PIK3CG, SLC6A4, TGFB1, and PTGS2) related to the anti-inflammatory effects of *F. rhynchophylla* on inflammatory dermatosis. In addition, all six compounds stably bound PTGS2. This study describes the anti-inflammatory effects of EEFR on DNFB-induced CD in mice. According to the results of this study, EEFR effectively prevented epidermal hyperplasia and hyperkeratosis in DNFB-treated mice by inhibiting the production of TNF-α, IFN-γ, IL-6, and MCP-1, and thus, ameliorated scaling, erythema, excoriation, and skin thickening. Finally, EEFR inhibited COX-2 expression and the activation of the NF-κB and JNK pathways in RAW 264.7 cells. Taken together, our findings indicate that *F. rhynchophylla* bark has potential use as a corticosteroid alternative for the treatment of inflammatory dermatoses and that its anti-inflammatory effects are associated with the suppressions of pro-inflammatory cytokine and chemokine levels and the NF-κB and JNK pathways.

## Figures and Tables

**Figure 1 ijms-24-06091-f001:**
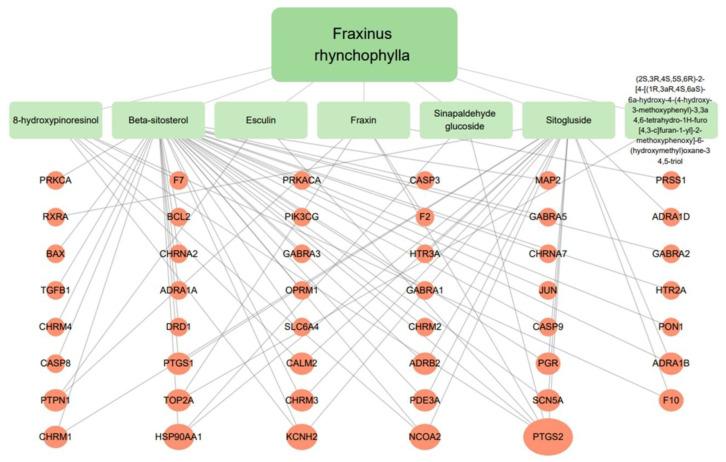
Seven active compounds and target genes in *F. rhynchophylla*.

**Figure 2 ijms-24-06091-f002:**
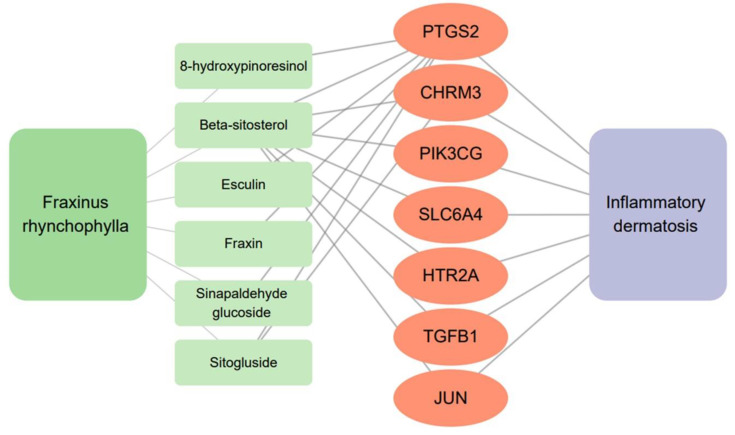
Six active compounds with targets associated with inflammatory dermatosis.

**Figure 3 ijms-24-06091-f003:**
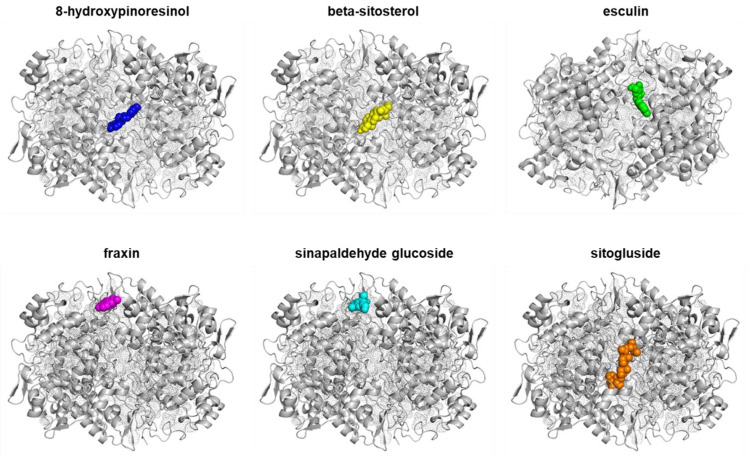
Binding modes of six active compounds and COX-2 Macromolecule (grey cartoon) means COX-2.

**Figure 4 ijms-24-06091-f004:**
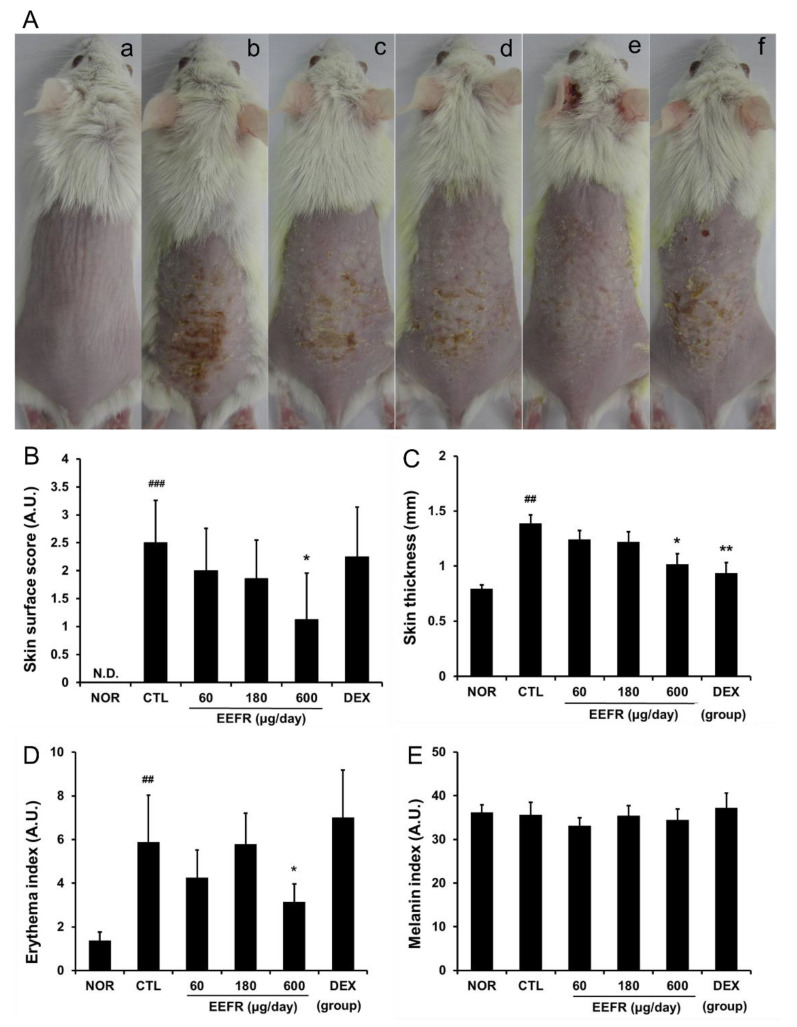
Effects of EEFR on skin lesions, thickness, and color in CD mice. Skin lesions were observed using a digital camera. (**a**), Treatment-naïve (NOR); (**b**), CD control (CTL); (**c**), 60 µg/day EEFR; (**d**), 180 µg/day EEFR; (**e**), 600 µg/day EEFR; and (**f**), 150 μg/day DEX (**A**). Skin surface scores were used to assess symptom severities (**B**). Weights of 5 mm diameter skin samples were measured using a micro-balance (**C**). Erythema and melanin indices were determined using a dermo-spectrophotometer (**D**,**E**). EEFR, ethanolic extract of *Fraxinus rhynchophylla* bark; DEX, dexamethasone. Results are presented as means ± standard deviation (SD)s. ^##^
*p* < 0.01 and ^###^
*p* < 0.001 vs. NOR; * *p* < 0.05 and ** *p* < 0.01 vs. CTL.

**Figure 5 ijms-24-06091-f005:**
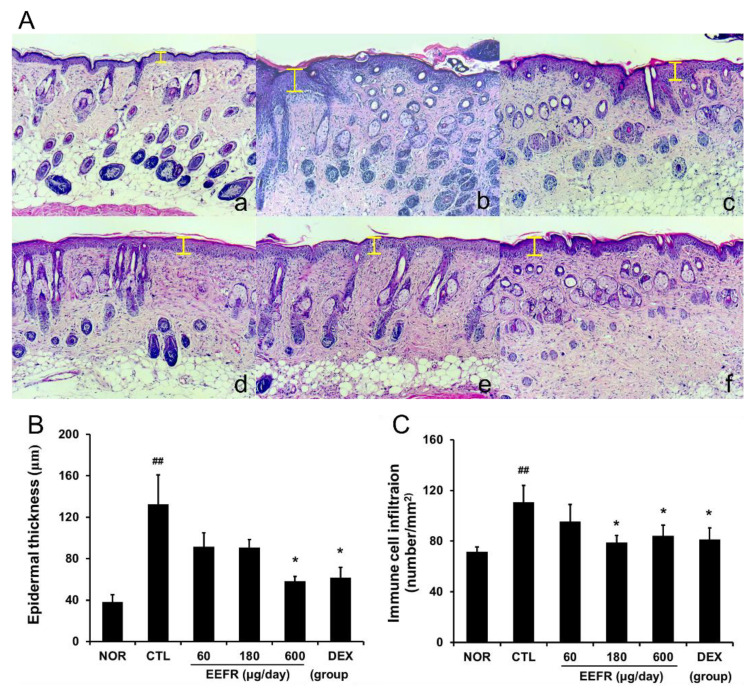
Effects of EEFR on histopathological abnormalities in inflamed tissues. The sequence is the same as in Figure 4A. The yellow bars indicate epidermal thicknesses (original magnification ×100) (**A**). Histogram showing mean epidermal thicknesses (**B**) and numbers of infiltrating immune cells (**C**) in the six study groups. Abbreviations are as defined in Figure 4. Results are presented as means ± SDs. ^##^
*p* < 0.01 vs. NOR; * *p* < 0.05 vs. CTL.

**Figure 6 ijms-24-06091-f006:**
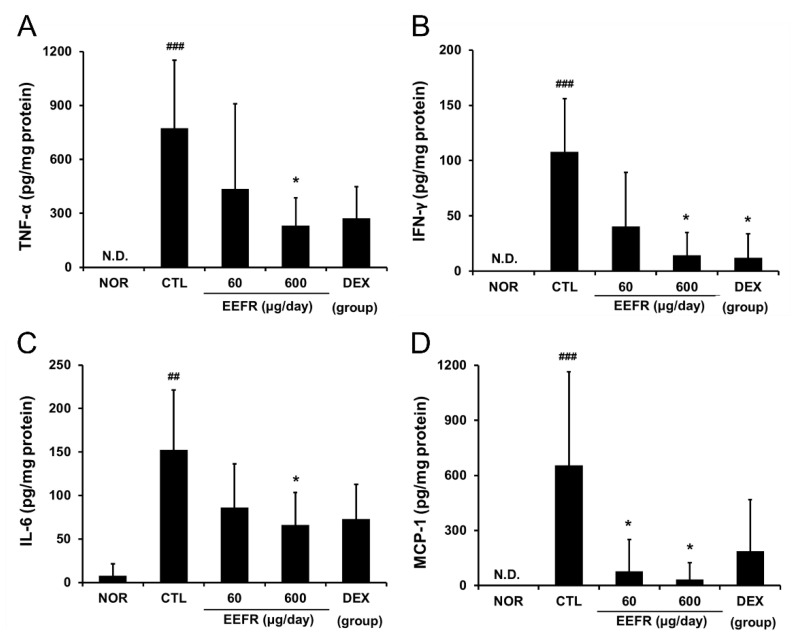
Effects of EEFR on cytokines and chemokines in inflamed tissues. Levels of cytokines and chemokines were evaluated using the cytometric bead array (CBA) method. (**A**) TNF-α; (**B**) IFN-γ; (**C**) IL-6; (**D**) MCP-1. N.D. means not detectable; abbreviations are as defined in Figure 4. Results are presented as means ± SDs. ^##^
*p* < 0.01 and ^###^
*p* < 0.001 vs. NOR; * *p* < 0.05 vs. CTL.

**Figure 7 ijms-24-06091-f007:**
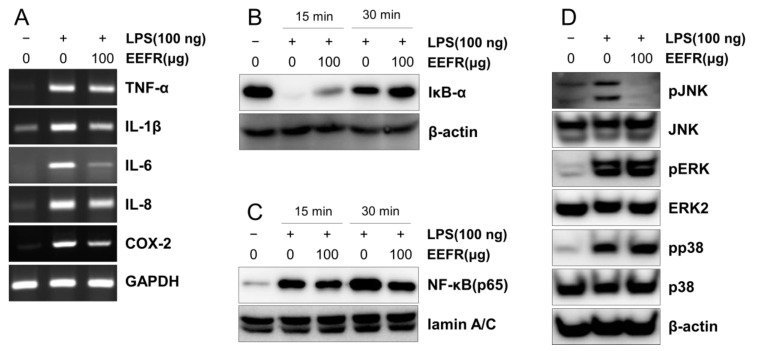
Effects of EEFR on LPS-induced increases on the expression of pro-inflammatory cytokines and COX-2, and on the NF-κB and MAPK signalling pathways in RAW 264.7 cells. Pro-inflammatory cytokine, chemokine, and COX-2 expressions were assessed with RT-PCR (**A**). Levels of IκB-α in cytoplasm (**B**), p65 in nuclear fractions (**C**), and MAPK signalling molecules (**D**) were determined using Western blots.

**Table 1 ijms-24-06091-t001:** Active compounds of *F. rhynchophylla*.

Compound Name	PubChem CID	Structure	Molecular Weight	OB (%)	DL
(2S,3R,4S,5S,6R)-2-[4-[(3R,3aS,6S,6aR)-3a-hydroxy-6-(4-hydroxy-3-methoxyphenyl)-3,4,6,6a-tetrahydro-1H-furo[3,4-c]furan-3-yl]-2-methoxyphenoxy]-6-(hydroxymethyl)oxane-3,4,5-triol	14756314	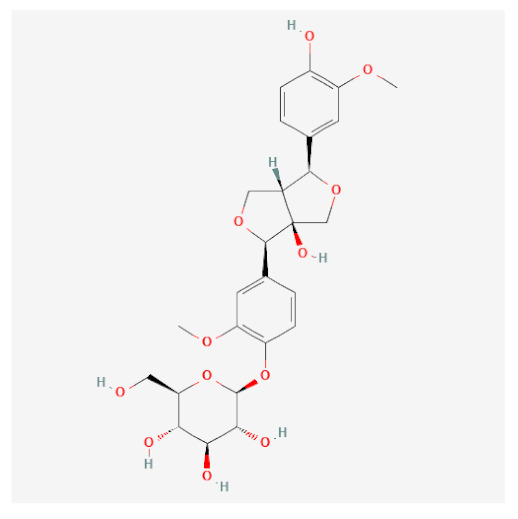	536.5	28.54	0.86
8-Hydroxypinoresinol	3010930	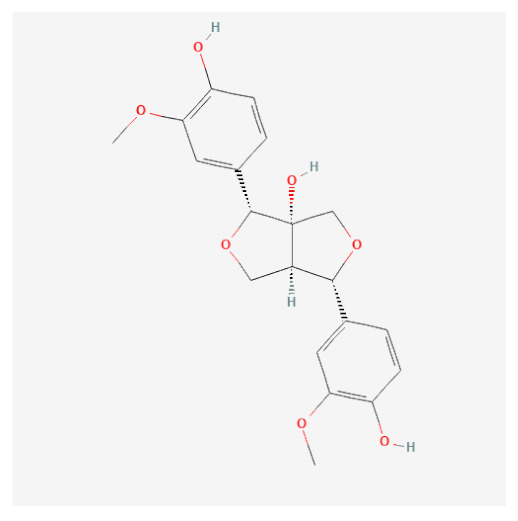	374.40	92.43	0.55
Beta-sitosterol	222284	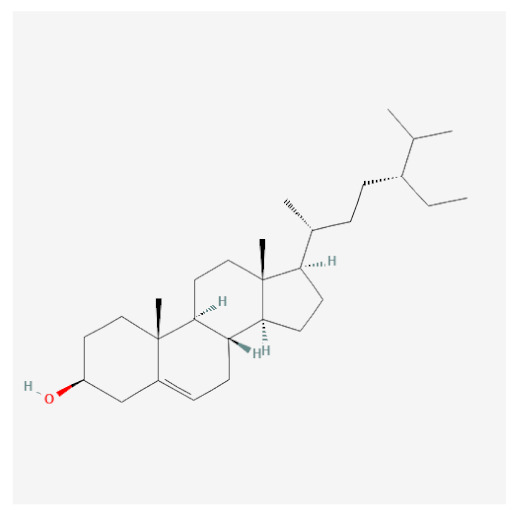	414.70	36.91	0.75
Esculin	5281417	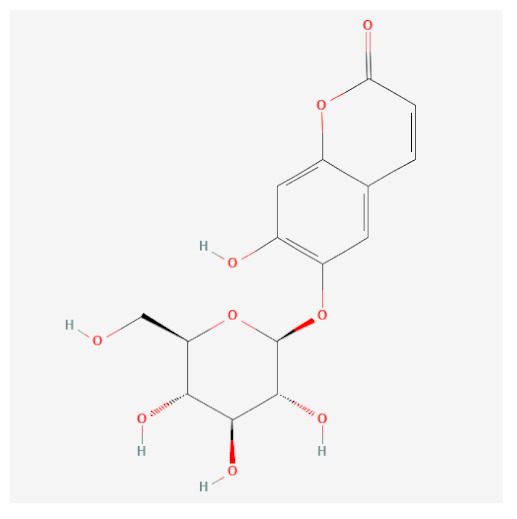	340.28	20.43	0.36
Fraxin	5273568	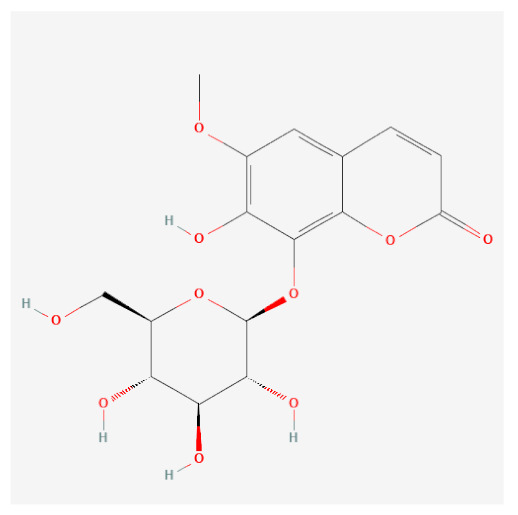	370.31	36.76	0.42
Sinapaldehyde glucoside	25791064	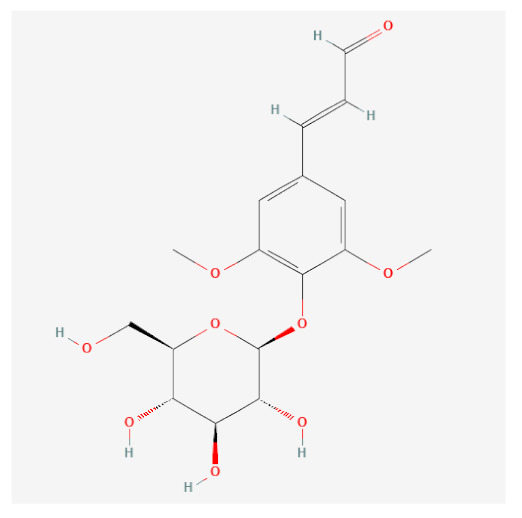	370.40	20.91	0.33
Sitogluside	5742590	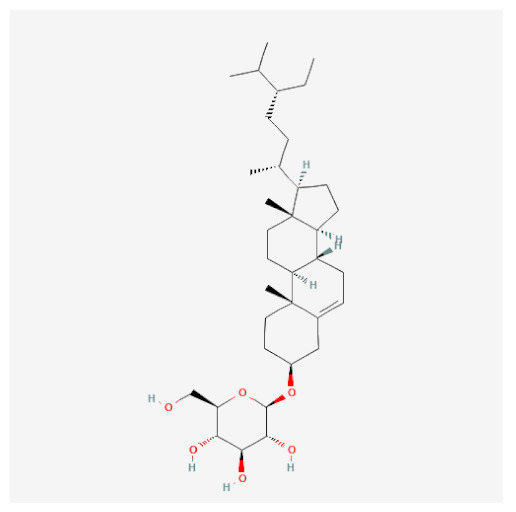	576.80	20.63	0.62

OB means oral bioavailability and DL means drug likeness.

**Table 2 ijms-24-06091-t002:** Binding affinities for interactions between PTGS2 (COX-2) and the six active compounds.

Compounds of F. rhynchophylla	Target (PDB-ID)	Affinity (kcal/mol)
8-Hydroxypinoresinol	PTGS2 (5f19)	−7.865
Beta-sitosterol		−8.251
Esculin		−7.451
Fraxin		−7.741
Sinapaldehyde glucoside		−6.538
Sitogluside		−8.654

PDB means protein data bank.

## Data Availability

The data presented in this study are available upon request from the corresponding author.

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
