# Peer review of "The Related Mechanisms Predicted through Network-Based Pharmacological Analysis and the Anti-Inflammatory Effects of Fraxinus rhynchophylla Hance Bark on Contact Dermatitis in Mice"

_ijms, 2023, doi:10.3390/ijms24076091_

Round 1
Reviewer 1 Report
The manuscript submitted for publication deals with the evaluation of the mechanisms involved in the anti-inflammatory effects of Fraxinus rhynchophylla Hance bark on contact dermatitis (CD) in mice.
The selected topic is pertinent, and the manuscript’s structure and content are convincing. Besides that, it encompasses a large number of experimental assays performed. Furthermore, the paper is in general well-written.
I suggest that this manuscript should be accepted for publication after the suggested revisions.
Some of the points are as below:
- Replace “ethanol extract” with “ethanolic extract”;
- Check all the abbreviations and put them in parenthesis after providing the corresponding full name;
- In the second paragraph of the Introduction, give more details about the CD and also about the standard treatment;
- Lines 76-77: “Clinical symptoms of CD include itching, erythema, eczema, psoriasis, and blister formation [7]”. Psoriasis is related to CD??
- The different subsections of the results must be rewritten, as it is presented they are a summary of the achieved results;
- Lines 226-227 – “.. in the present study, repeated application of EEFR ameliorated scaling, pigmentation, erythema, and skin thickness …” According to the results, it seems that EEFR didn´t exert an effect on the pigmentation. Please check it.
- The last paragraph of the Discussion section should be moved to the Conclusions section.
- Line 278 (“therapeutic efficacy”) and line 382 (“therapeutic effect”) – Replace “therapeutic” as it is used for drugs.
With thanks and best wishes,
The reviewer
Author Response
All of your comments have been carefully considered and the manuscript has been revised accordingly. The corrections to your comments are attached as word documents below.

Reviewer 2 Report
The current manuscript presented the active compounds in F. rhynchophylla and the analysis of the potential target genes. Additionally, an ethanol extract of F. rhynchophylla displays excellent efficacy in a mouse model of contact dermatitis (CD). In vitro cellular studies shows it also suppressed inflammatory cytokine levels, COX-2, the nuclear translocation of 32 NF-B (p65) and the activation of c-Jun N-terminal kinases (JNK). While the data look interesting and could be relevant in developing therapeutic strategies for CD, the manuscript is expected to address the following major issues:
1) The primary cells involved in CD are CD4 T cells, CD8 T cells, and epidermal keratinocytes. In the in vitro studies, the Raw264.7 cells were used to demonstrate the potential mechanisms, which could be pathologically irrelevant. An intensive discussion should be included in the discussion to address this issue or limitation.
2) Given the quantification nature and variability in Figure 4, the one-way ANOVA test is not appropriate for the statistical analysis. Instead, the nonparametric methods should be chosen for the analysis.
3) In Figure 4B, the steroids did not show a significant effect. This is unusual, as steroids have proved to be extremely effective in mouse models and human patients with CD. A brief discussion is necessary to explain this result.
4) In Figure 7, it is unclear whether these results are representative that must be repeated. A quantification or semi-quantification data should be helpful. ERC1/2 rather than ERC2 should be chosen for the protein load control in Panel D.
Author Response

(The authors gave the same response as above.)

Round 2
Reviewer 2 Report
The manuscript has been dramatically improved after revision. However, the manuscript has to be edited due to a number of minor issues. Here are some examples:
Line 86: Topical corticosteroid is the treatment of choice for inflammatory skin diseases. This description is not appropriate and has to be rephrased.
Line 252: “skin inflammatory disorders” has to be changed to “inflammatory skin disorders”.
Line 305: The abbreviation use of DEX has to be placed in Line 297 where it appears first.
Author Response

(The authors gave the same response as above.)
